

# Comparative postoperative prognosis of ceramic-on-ceramic and ceramic-on-polyethylene for total hip arthroplasty: an updated systematic review and meta-analysis

Tingyu Wu[1], Yaping Jiang[2], Weipeng Shi[1], Yingzhen Wang[1] and Tao Li[1]

[1] Department of Joint Surgery, The Affiliated Hospital of Qingdao University, Qingdao, Shandong, China
[2] Department of Oral Implantology, The Affiliated Hospital of Qingdao University, Qingdao, Shandong, China

## ABSTRACT

**Objective.** To compare the clinical outcomes between ceramic-on-ceramic (CoC) and ceramic-on-polyethylene (CoP) bearing surfaces in patients undergoing total hip arthroplasty (THA) through a pooled analysis and evidence update.

**Methods.** We performed a systematic literature search using PubMed, Embase, Cochrane Library and Web of Science up to March 2023 for studies that compared the bearing surfaces of CoC and CoP in patients undergoing THA. The primary outcomes were the incidence of common postoperative complications and the rate of postoperative revision. The secondary outcome was the Harris Hip Score.

**Results.** A total of 10 eligible studies involving 1,946 patients (1.192 CoC-THA *versus* 906 CoP-THA) were included in the evidence synthesis. Pooled analysis showed no significant difference in the rates of common postoperative complications (dislocation, deep vein thrombosis, infection, wear debris or osteolysis) and of revision. After eliminating heterogeneity, the postoperative Harris Hip Score was higher in the CoC group than in the CoP group. However, the strength of evidence was moderate for the Harris Hip Score.

**Conclusion.** CoC articulations are more commonly used in younger, healthier, and more active patients. While the performance of conventional polyethylene is indeed inferior to highly cross-linked polyethylene, there is currently a lack of sufficient research comparing the outcomes between highly cross-linked polyethylene and CoC bearing surfaces. This area should be a focal point for future research, and it is hoped that more relevant articles will emerge. Given the limited number of studies included, the heterogeneity and potential bias of those included in the analysis, orthopaedic surgeons should select a THA material based on their experience and patient-specific factors, and large multicentre clinical trials with >15 years of follow-up are needed to provide more evidence on the optimal bearing surface for initial THA.

Corresponding author
Tao Li, qdult@qdu.edu.cn

## INTRODUCTION

After non-operative treatments have failed, total hip arthroplasty (THA) is often a recommended option for most young patients with osteoarthritis or other degenerative diseases of the hip. In fact, it is projected that there will be 635,000 THAs performed in the United States over the next decade (*Sloan, Premkumar & Sheth, 2018*). Although the survival rate of patients undergoing THA has significantly improved in many countries, such as China and the United Kingdom, many young patients need revision surgery due to wear, aseptic loosening, and other problems (*Buddhdev et al., 2020*; *Zeng et al., 2019*). Are these reasons for revision related to the type of prosthesis? The choice of implant combination in primary THA has long been controversial, it is still unclear which of these articulations is superior in the long or very long term.

At present, ceramic-on-ceramic (CoC) and ceramic-on-polyethylene (CoP) bearings are the most commonly used in clinical work during THA (*Tsikandylakis et al., 2020*), but which should be used in a given clinical situation is unclear. Research studies have shown that CoC bearings for THA can lower the risk of wear debris, aseptic loosening, and lead to high 10-year survival rates in patients under 60 years old (*El-Desouky II, Helal & Mansour, 2021*; *Niu et al., 2022*). On the other hand, CoP bearings has been found to reduce the risk of postoperative periprosthetic joint infection (*Chisari et al., 2022*), while using ceramic-on-highly-crosslinked-polyethylene (HXLPE) resulted in less wear debris and no increased risk of postoperative complications (*Almaawi et al., 2021*).

*Dong et al. (2015)* were the first to evaluate CoC and CoP bearings for THA. In addition, two meta-analyses compared postoperative complication and revision rates between the CoC-THA and CoP-THA groups, but neither evaluated the postoperative prognosis or the optimum bearing surface for THA (*Shang & Fang, 2021*; *Van Loon et al., 2022*). Two novel studies on this matter were published in 2021 and 2022 (*Giuseppe et al., 2022*; *Van Loon et al., 2021*). This meta-analysis compared the results of the two bearing surfaces after THA, objectively (complications and revision) and subjectively (Harris Hip Score). We also reviewed the latest evidence, to provide guidance in the preoperative selection of CoC bearings or CoP bearings.

## METHODS

### Protocol and registration

This systematic review and meta-analysis followed a preprint registered in the International Prospective Register of Systematic Reviews (PROSPERO; CRD42023400537) and was designed and conducted according to the guidelines in the Cochrane Handbook.

### Search strategy and data sources

We conducted a meta-analysis of all studies that compared CoC and CoP for THA, identified by searching PubMed, Embase, Cochrane Library, and Web of Science up to March 2023. References were managed using Endnote X9 software (Clarivate Analytics). In addition, relevant reviews and the references of selected articles were examined for potentially relevant trials.

## Eligibility criteria

The inclusion criteria were as follows: (1) prospective randomised studies; (2) retrospective randomised studies (3) randomised controlled trials (RCTs) (3) studies of patients undergoing primary THA; (3) studies comparing COC-THA and COP-THA; (4) studies reporting complications or revisions or Harris Hip Score; (5) studies in which the mean age of the patients was >40 years; (6) studies published in the English language.

The following studies were excluded: (1) non-human studies; (2) non-original studies (letters, reviews, editorials); (3) studies with patients who had no THA; (4) studies of revision surgery; (5) interim follow-up of the same study; and (6) studies without available data.

## Study selection and data extraction

After removing duplicates, each article was screened independently by two reviewers who were blind to the journal, author, institution at which the study was performed and the date of publication, to identify potentially eligible studies and relevant clinical trials. One reviewer (TYW) was responsible for extracting the data, and another (YPJ) checked the data for accuracy against the source material. The final eligibility of the retrieved full-text articles for inclusion was assessed independently by two reviewers. Differences of opinion were resolved by discussion, and if no agreement was reached, the third reviewer (TL) made the final decision.

## Quality assessment

The quality of included studies was evaluated using the Cochrane risk of bias assessment tool. The standards implemented by the Oxford Centre for Evidence-based Medicine were applied to assess the level of evidence. Two investigators independently evaluated the quality and level of evidence for eligible studies. An arbiter was consulted to reconcile any disagreements.

## Statistical analysis

Statistical analysis was performed with Review Manager 5.4 (Cochrane Collaboration, Oxford, UK) and Stata 12.0 (StataCorp LP, College Station, Texas) software. The weighted mean difference (WMD) and 95% CI were calculated for the continuous outcome (Harris Hip Score). The medians and interquartile ranges of continuous data were converted to means and standard deviations. The results of binary variables (dislocation, fracture, deep vein thrombosis, infection, wear debris or osteolysis and revision) are expressed as odds ratios (OR) with 95% confidence intervals (CI). For meta-analyses, the Cochrane Q $p$-value and $I^2$ statistic were applied to assess heterogeneity. A $p < 0.05$ or $I^2 > 50\%$ indicated significant heterogeneity; in such cases, a random-effects model was used to estimate the combined WMD or OR. Otherwise, a fixed-effects model was used. We performed one-way sensitivity analyses to evaluate the effects of included studies on the combined results for outcomes with significant heterogeneity. A value of $p < 0.05$ was considered indicative of statistical significance.

## RESULTS

### Study selection and characteristics

Figure 1 outlines study selection and reasons for exclusion. A total of 3,640 relevant articles in PubMed ($n = 888$), Embase ($n = 1,272$), Cochrane Library ($n = 99$), and Web of Science ($n = 1,381$) were identified. After removing repeated studies, and reading titles and abstracts, 10 full-text articles involving 1,946 patients and 2,098 THA surgeries were included in the pooled analysis. These articles comprised seven prospective randomised studies (*Van Loon et al., 2021*; *Amanatullah et al., 2011*; *Atrey et al., 2018*; *Bal et al., 2005*; *Feng et al., 2019*; *Lewis et al., 2010*; *Ochs et al., 2007*), two retrospective studies (*Giuseppe et al., 2022*; *Epinette & Manley, 2014*) and one RCT (*Beaupre, Al-Houkail & Johnston, 2016*). All studies were in English and were published between 2005 and 2022. The mean follow-up period was 2–15 years. HXLPE liners were used in two of the studies (*Feng et al., 2019*; *Epinette & Manley, 2014*), and conventional polyethylene liners in the others (*Giuseppe et al., 2022*; *Van Loon et al., 2021*; *Amanatullah et al., 2011*; *Atrey et al., 2018*; *Lewis et al., 2010*; *Ochs et al., 2007*; *Beaupre, Al-Houkail & Johnston, 2016*). All the articles presented baseline age, sex, and body mass index values. The characteristics of the studies are listed in Table 1. We did not assess publication bias because <10 studies were included in the required observation measures.

### Demographic characteristics

There was no significant difference between the two groups in terms of gender (female/total, OR: 0.90; 95% CI [0.74–1.09] $p = 0.27$), BMI (WMD: 0.04; 95% CI [−1.20 to 1.29]; $p = 0.94$). However, the two groups were significantly different in baseline age (WMD: −4.07; 95% CI [−5.93 to −2.20]; $p < 0.0001$) (Table 2). The mean age of patients in the CoP-THA group was 4 years older than in the CoC-THA group (*Giuseppe et al., 2022*; *Van Loon et al., 2021*; *Amanatullah et al., 2011*; *Atrey et al., 2018*; *Bal et al., 2005*; *Feng et al., 2019*; *Lewis et al., 2010*; *Ochs et al., 2007*; *Epinette & Manley, 2014*; *Beaupre, Manolescu & Johnston, 2013*).

### Risk of bias

Overall, the risk of bias was low across the trials. All trials used allocation concealment, followed a blinded design, and no selective reporting occurred. One trial (*Bal et al., 2005*) did not describe the sources of the exposed and non-exposed groups, and the exposed group in one trial (*Beaupre, Al-Houkail & Johnston, 2016*) was not representative, all of whom were volunteers. The rate of loss to follow-up was >10% in five trials (*Amanatullah et al., 2011*; *Atrey et al., 2018*; *Bal et al., 2005*; *Ochs et al., 2007*; *Epinette & Manley, 2014*), and loss to follow-up was not clearly described in one trial (*Van Loon et al., 2021*). The study groups of four trials (*Van Loon et al., 2021*; *Amanatullah et al., 2011*; *Bal et al., 2005*; *Feng et al., 2019*) were significantly different in age from the control group and were not comparable in age, so other biases may have been introduced. Figures 2 and 3 summarise the risks of bias of the included studies.

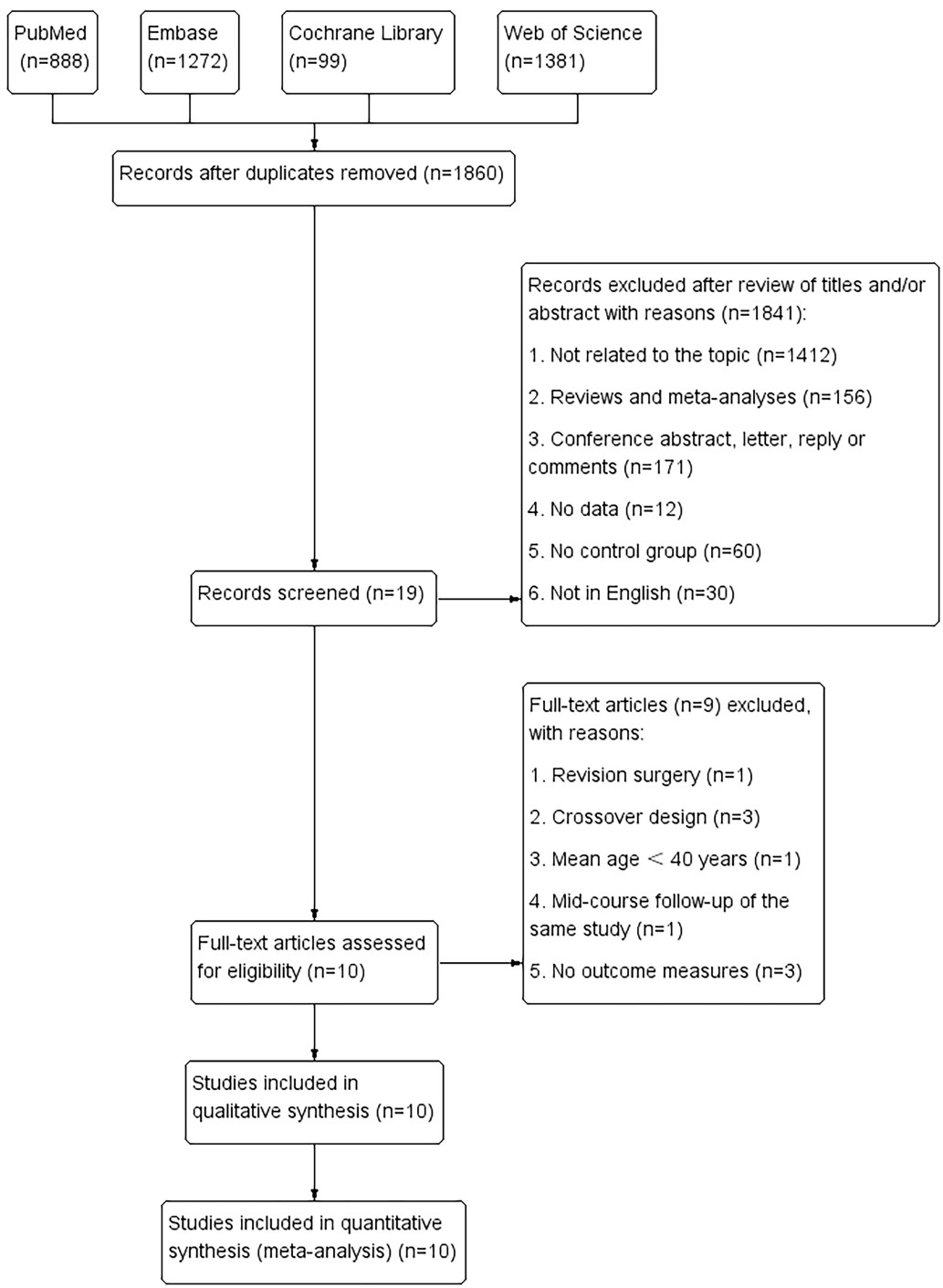

**Figure 1  Flowchart of the systematic search and selection process.**

**Table 1 Baseline characteristics of include studies.**

| Authors | Study period | Region | Study design | Follow-up (y) | Patientsr (hips, n) | |
|---|---|---|---|---|---|---|
| | | | | | CoC | CoP |
| *Amanatullah et al. (2011)* | 1999–2001 | USA | Prospective | 5 | 166 (196) | 146 (161) |
| *Atrey et al. (2018)* | 1997–1999 | Canada | Prospective | 15 | 29 (29) | 28 (29) |
| *Bal et al. (2005)* | 1998–2001 | USA | Prospective | 2 | 238 (250) | 241 (250) |
| *Beaupre, Al-Houkail & Johnston (2016)* | 1998–2003 | Canada | RCT | 10 | 48 (48) | 44 (44) |
| *Epinette & Manley (2014)* | 1997–2002 | France | Retrospective | 10 | 412 (447) | 216 (228) |
| *Feng et al. (2019)* | 2009–2012 | China | Prospective | 7 | 71 (93) | 62 (77) |
| *Giuseppe et al. (2022)* | 2005–2008 | Italy | Retrospective | 15 | 43 (43) | 43 (43) |
| *Lewis et al. (2010)* | 1997–1999 | Canada | Prospective | 8 | NA (30) | NA (26) |
| *Ochs et al. (2007)* | 1997–1999 | Germany | Prospective | 8.1 | 22 (22) | 21 (21) |
| *Van Loon et al. (2021)* | 2003–2004 | Netherlands | Prospective | 10 | 34 (34) | 27 (27) |

| Author | Mean age (SD) | | Material design | Female (%) | | BMI (SD) | |
|---|---|---|---|---|---|---|---|
| | CoC | CoP | | CoC | CoP | CoC | CoP |
| *Amanatullah et al. (2011)* | 50.4 (12.8) | 54.7 (12.9) | A-on-A VS. A-on-PE | 36.1 | 42.5 | 29.6 (12.4) | 29.6 (12.4) |
| *Atrey et al. (2018)* | 50.4 (12.8) | 54.7 (12.9) | A-on-A VS. A-on-PE | 41.4 | 46.4 | 26.7(6.6) | 28.2(5.2) |
| *Bal et al. (2005)* | 54.97 (14.7) | 60.93 (12.8) | A-on-A VS. A-on-PE | 47.1 | 55.2 | NA | NA |
| *Beaupre, Al-Houkail & Johnston (2016)* | 51.3 (6.9) | 53.6 (6.5) | A-on-A VS. A-on-PE | 45.8 | 45.4 | NA | NA |
| *Epinette & Manley (2014)* | 68.04 (9.7) | 68.66 (10.0) | A-on-A VS. A-on-HXLPE | 73.3 | 69.4 | 27.4 (4.5) | 28.14 (4.93) |
| *Feng et al. (2019)* | 51.21 (9.6) | 58.77 (9.3) | A-on-A VS. A-on-HXLPE | 56.3 | 53.2 | 25.12(1.98) | 23.27(1.98) |
| *Giuseppe et al. (2022)* | 63.4 (6.5) | 67.8 (11.0) | A-on-A VS. A-on-PE | 46.5 | 51.2 | 27 (3.1) | 25.9 (3.3) |
| *Lewis et al. (2010)* | 41.5 (8.9) | 42.8 (6.9) | A-on-A VS. A-on-PE | NA | NA | 26.7 (6.6) | 28.2 (5.2) |
| *Ochs et al. (2007)* | 64.4 (7.8) | 69.2 (7.2) | A-on-A VS. A-on-PE | 31.8 | 33.3 | NA | NA |
| *Van Loon et al. (2021)* | 55.7 (8.5) | 64.2 (5.3) | A-on-A VS. A-on-PE | 64.7 | 77.8 | 26.9 (4.1) | 27.6 (4.1) |

Notes.

A-on-A, Alumina-on-Alumina; A-on-PE, Alumina-on-Polyethylene; A-on-HXLPE, Alumina-on- Highly-Crosslinked-Polyethylene; NA, not available; BMI, body mass index.

## Primary outcome: rates of common postoperative complications and revision surgery

### Dislocation rate

Postoperative dislocations were reported in seven of the 10 studies, involving 1,742 patients (1,086 CoC-THA *versus* 807 CoP-THA) (*Amanatullah et al., 2011*; *Bal et al., 2005*; *Feng et al., 2019*; *Lewis et al., 2010*; *Ochs et al., 2007*; *Epinette & Manley, 2014*; *Beaupre, Al-Houkail & Johnston, 2016*). There was no significant difference between the two groups (OR: 0.85; 95% CI [0.51–1.42]; $p = 0.54$), and no significant heterogeneity ($I^2 = 0\%$, $p = 0.77$) (Fig. 4A).

### Deep vein thrombosis rate

Three studies with 579 patients (468 CoC-THA *vs.* 432 CoP-THA) were included in the analysis of the postoperative deep vein thrombosis rate (*Amanatullah et al., 2011*; *Bal et al., 2005*; *Ochs et al., 2007*). Evidence synthesis revealed a similar deep vein thrombosis rate in the two groups (OR: 1.22; 95% CI [0.44–3.43]; $p = 0.70$) without significant heterogeneity ($I^2 = 0\%$, $p = 0.83$) (Fig. 4B).

Wu et al. (2024), *PeerJ*, DOI 10.7717/peerj.18139

**Table 2   Demographics characteristics of included studies.**

| Outcomes | Studies | No. of patients | WMD or OR | 95% CI | p-value | Heterogeneity | | | |
|---|---|---|---|---|---|---|---|---|---|
| | | CoC-THA/CoP-THA | | | | Chi$^2$ | df | p-value | I$^2$ (%) |
| Age (years) | *Giuseppe et al. (2022)*; *Van Loon et al. (2021)*; *Amanatullah et al. (2011)*; *Atrey et al. (2018)*; *Bal et al. (2005)*; *Feng et al. (2019)*; *Lewis et al. (2010)*; *Ochs et al. (2007)*; *Epinette & Manley (2014)*; *Beaupre, Al-Houkail & Johnston (2016)* | 1,093/854 | −4.07 | [−5.93, −2.20] | <0.0001 | 33.56 | 9 | 0.0001 | 73 |
| Gender (female) | *Giuseppe et al. (2022)*; *Van Loon et al. (2021)*; *Amanatullah et al. (2011)*; *Atrey et al. (2018)*; *Bal et al. (2005)*; *Feng et al. (2019)*; *Ochs et al. (2007)*; *Epinette & Manley (2014)*; *Beaupre, Al-Houkail & Johnston (2016)* | 1,063/828 | 0.90 | [0.74, 1.09] | 0.27 | 5.86 | 7 | 0.56 | 0 |
| BMI (kg/m$^2$) | *Giuseppe et al. (2022)*; *Van Loon et al. (2021)*; *Amanatullah et al. (2011)*; *Atrey et al. (2018)*; *Feng et al. (2019)*; *Lewis et al. (2010)*; *Epinette & Manley (2014)* | 785/548 | 0.04 | [−1.20, 1.29] | 0.94 | 30.07 | 6 | <0.0001 | 80 |

**Notes.**

BMI, body mass index; WMD, weighted mean difference; OR, odds ratio; CI, confidence interval.

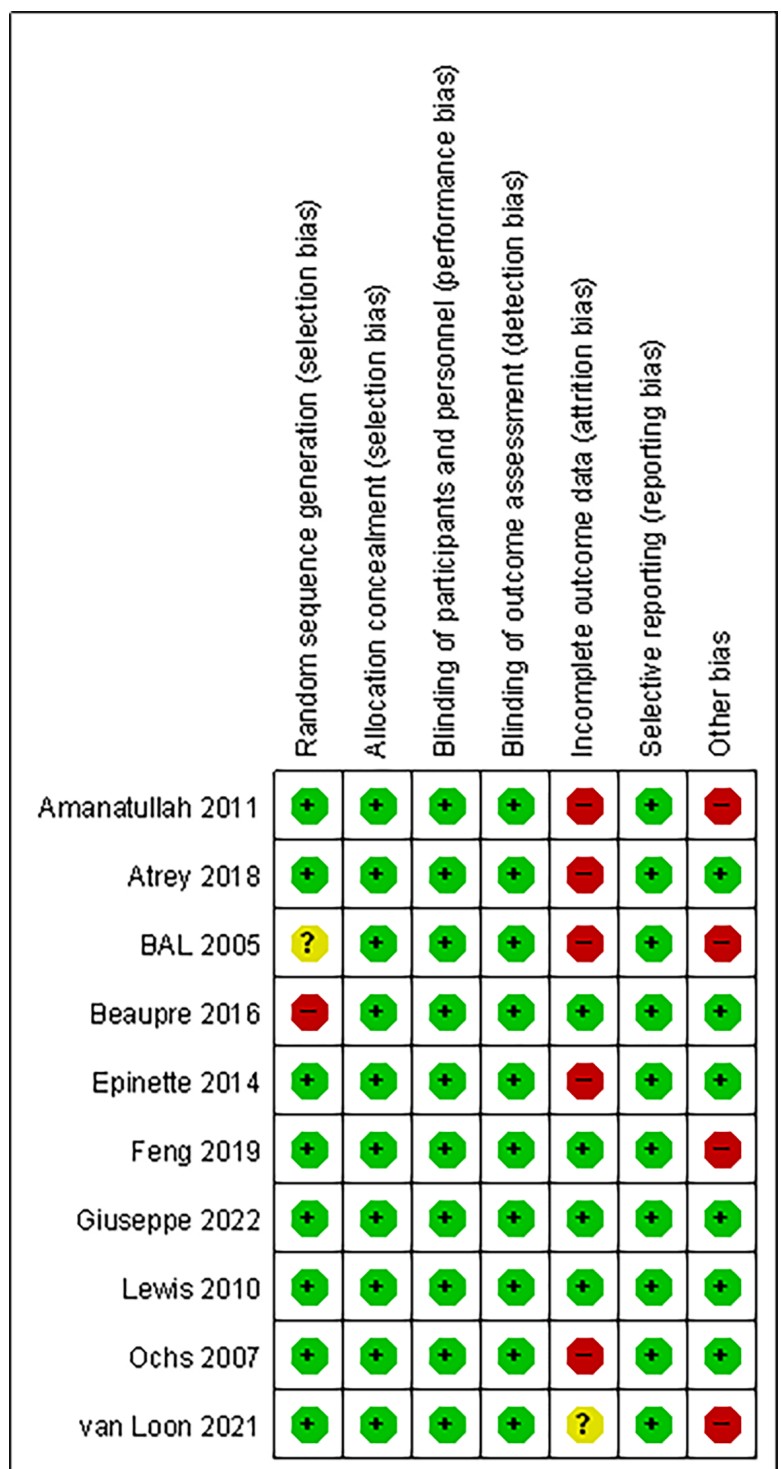

**Figure 2** Quality assessment of risk of bias summary in included studies.

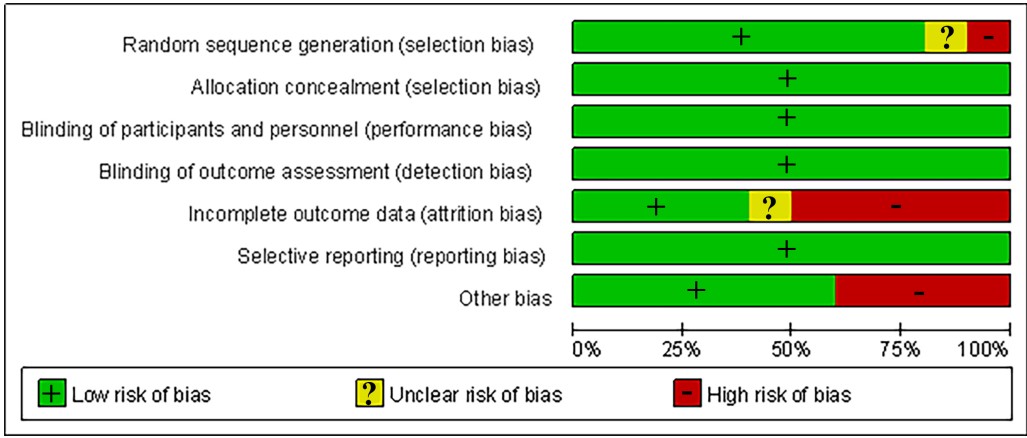

**Figure 3 Risk of bias graph.**

### Infection rate

Five of the ten studies provided the number of infection events after THA, one of which reported periprosthetic joint infection (*Van Loon et al., 2021*), two reported superficial infection and deep infection (*Amanatullah et al., 2011*; *Feng et al., 2019*) and two studies did not clearly describe the site of infection (*Atrey et al., 2018*; *Ochs et al., 2007*). A total of 606 patients were analysed (374 CoC-THA *vs.* 315 CoP-THA). The rate of infection was similar in the two groups (OR: 1.28; 95% CI [0.52–3.17]; $p = 0.59$), and no significant heterogeneity ($I^2 = 0\%$, $p = 0.80$) was detected (Fig. 4C).

### Wear debris or osteolysis rate

Three articles reported generation of much wear debris and subsequently osteolysis during postoperative follow-up, involving 455 patients (268 CoC-THA *vs.* 232 CoP-THA) (*Giuseppe et al., 2022*; *Amanatullah et al., 2011*; *Atrey et al., 2018*). The wear debris or osteolysis rates were non-significantly lower in the CoC-THA group compared to the CoP-THA group, but do approach significance with a $p$ value of 0.06 (OR: 0.41; 95% CI [0.16–1.05]; $p = 0.06$). No significant heterogeneity ($I^2 = 0\%$, $p = 0.85$) was detected (Fig. 4A). What I need to note here is that conventional polyethylene, rather than crosslinked polyethylene, was used in the three papers we included for the the section on wear debris.

## Revision surgery

Seven of the ten studies reported information on the revision surgery (*Van Loon et al., 2021*; *Amanatullah et al., 2011*; *Atrey et al., 2018*; *Bal et al., 2005*; *Ochs et al., 2007*; *Epinette & Manley, 2014*; *Beaupre, Al-Houkail & Johnston, 2016*). Revision surgery was performed in 949 patients (1,026 hips) in the CoC-THA group and 723 patients (760 hips) in the CoP-THA group. The mean follow-up duration was 2.0–15 years. Common causes included hip instability, loose components, wear and osteolysis, infection, recurrent or multiple dislocation and implant or periprosthetic fracture, all of which were common postoperative complications (*Jungwirth-Weinberger et al., 2022*; *Ledford et al., 2019*). A small number of patients experienced stem subsidence, implant tilt, and persisting pain of

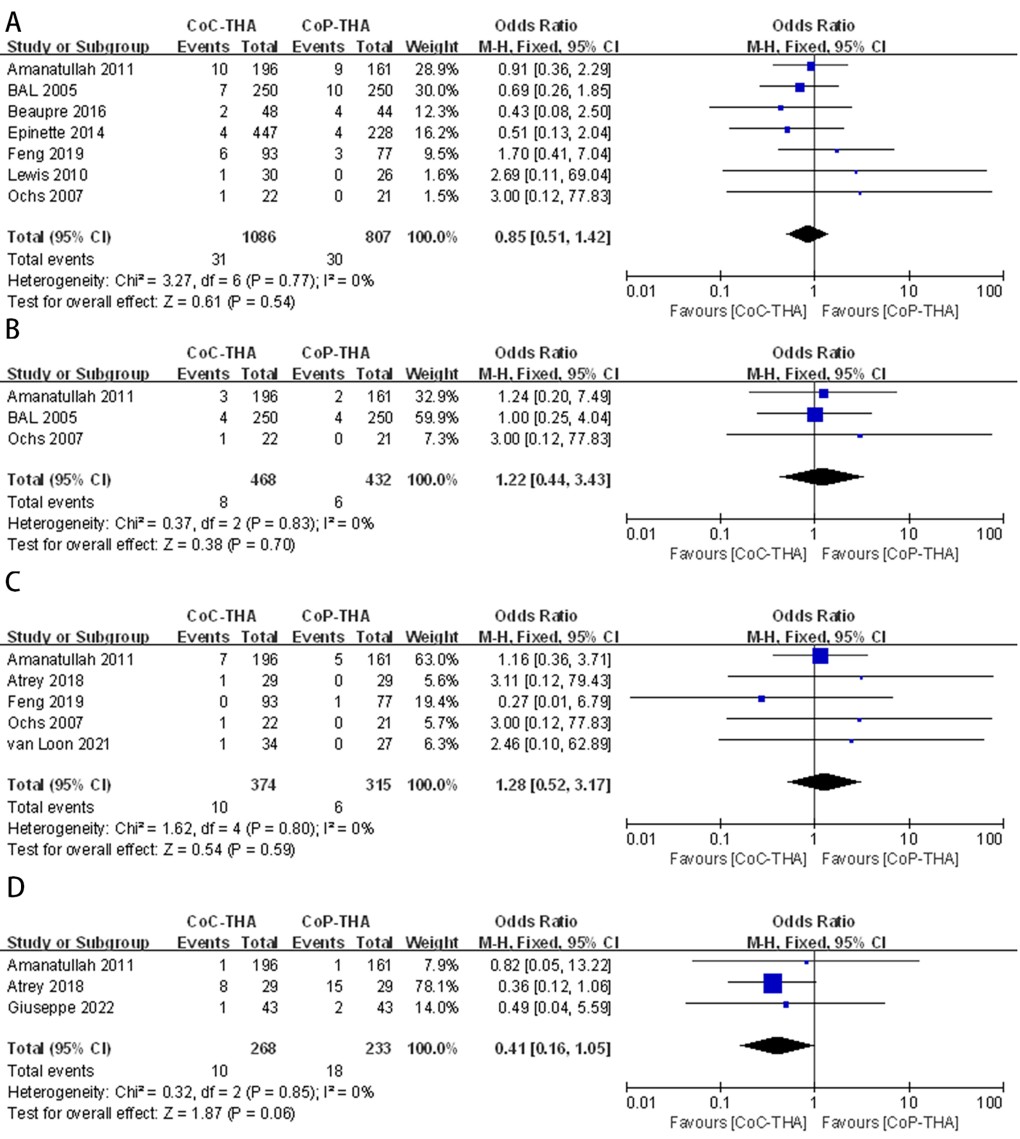

**Figure 4** Forest plots of postoperative complication outcomes: (A) dislocation rate, (B) deep venous thrombosis rate, (C) infection rate, (D) wear debris or osteolysis rate.

unknown cause after surgery, so revision surgery was performed. The revision surgery rate in the CoC-THA group was non-significantly lower (RR: 0.77; 95% CI [0.45–1.32]; $p = 0.35$). No significant heterogeneity ($I^2 = 38\%$, $p = 0.14$) was observed (Fig. 5).

## Secondary outcome: postoperative Harris Hip Score

Because few of the included studies evaluated hip function after surgery, there was insufficient information to compare hip function among implant combinations. The Harris Hip Score was used in six studies. Because one study did not differentiate between CoC-THA and CoP-THA scores (*Amanatullah et al., 2011*), and another did not clearly describe the standard deviation or range of scores (*Feng et al., 2019*), only four sets of data

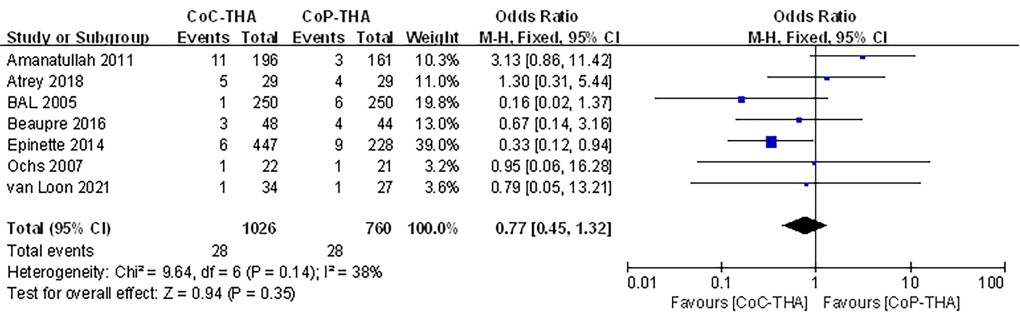

**Figure 5** Forest plots of revision surgery outcomes.

**Table 3** Preoperative and postoperative Harris hip score.

| Author | Patients (hips, n) | | Mean Harris hip score (SD) | | | |
|---|---|---|---|---|---|---|
| | CoC-THA | CoP-THA | CoC-THA | | CoP-THA | |
| | | | Preoperative | Postoperative | Preoperative | Postoperative |
| *Atrey et al. (2018)* | 29 (29) | 28 (29) | 50.3 (13.7) | 48.8 (19.9) | 94.6 (5.5) | 88.7 (10.5) |
| *Epinette & Manley (2014)* | 412 (447) | 216 (228) | 40.12 (10.6) | 44 (9.5) | 98.53 (2.96) | 98.29 (3.91) |
| *Feng et al. (2019)* | 71 (93) | 62 (77) | 47.9 (NA) | 40.4 (NA) | 89.6 (NA) | 86.7 (NA) |
| *Ochs et al. (2007)* | 22 (22) | 21 (21) | NA | NA | 91 (21.6) | 89 (12.1) |
| *Van Loon et al. (2021)* | 34 (34) | 27 (27) | 47.5 (13.4) | 50.2 (13.3) | 91.4 (17.0) | 87.3 (18.5) |

**Notes.**
NA, not available.

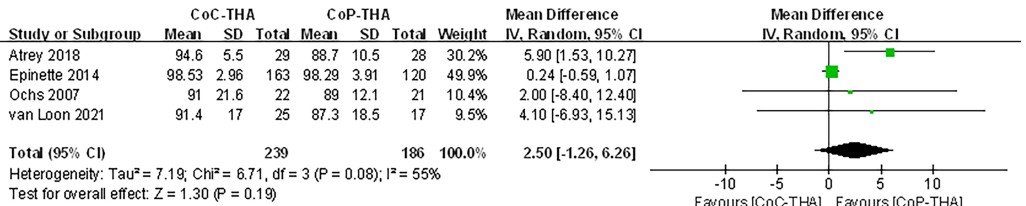

**Figure 6** Forest plots of postoperative Harris hip score.

were analysed, involving 420 patients (239 CoC-THA *vs.* 186 CoP-THA) (*Van Loon et al., 2021*; *Atrey et al., 2018*; *Ochs et al., 2007*; *Epinette & Manley, 2014*). The mean follow-up duration was 8.1–15 years. The Harris Hip Score improved significantly in both groups compared to that preoperatively (Table 3), it was non-significantly higher in the CoC-THA group (WMD: 2.50; 95% CI [−1.26 to 6.26]; $p = 0.19$). However, statistically significant heterogeneity was observed ($I^2 = 55\%$, $p = 0.08$) (Fig. 6).

Because of the >50% heterogeneity in postoperative Harris Hip Score, we conducted one-way sensitivity analyses to evaluate the influence of each individual study on the combined WMD by removing the studies one-by-one. Exclusion of *Epinette & Manley (2014)* eliminated the heterogeneity of the Harris Hip Score ($I^2 = 0\%$, $p = 0.78$) (Fig. 7) and the postoperative Harris Hip Score was significantly higher in the CoC-THA group

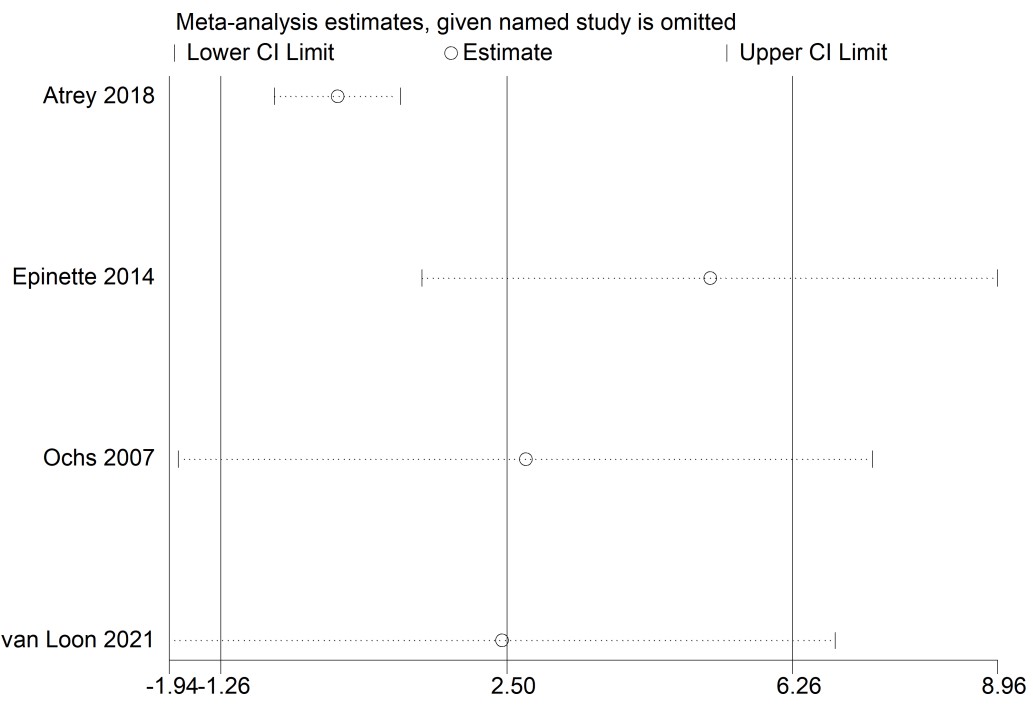

**Figure 7** Sensitivity analysis of postoperative Harris Hip Score.

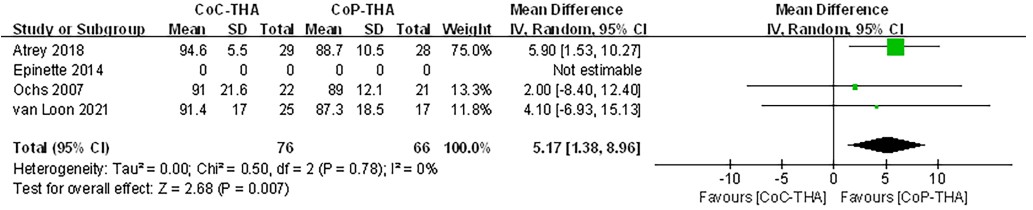

**Figure 8** Forest plots of postoperative Harris Hip Score after excluding one study.

than in the CoP-THA group (WMD: 5.17; 95% CI [1.38–8.96]; $p = 0.007$) (Fig. 8), suggesting that this study accounted for most of the heterogeneity.

## DISCUSSION

In this meta-analysis based on seven prospective randomised studies, two retrospective randomised studies and one RCT (1,946 patients; 1,192 CoC-THA *vs.* 906 CoP-THA), we found no evidence of a significant difference between CoC-THA and CoP-THA in objective indicators (incidence of common postoperative complications and the rate of postoperative revision) and a subjective indicator (Harris Hip Score) of the postoperative prognosis. This is likely to be because both bearing couples achieved good clinical results in the long-term follow-up.

Although there was no significant difference in dislocation rate between the CoC-THA and CoP-THA groups, dislocation was associated with liners and femoral heads of different sizes (*Bader et al., 2004*), which were not controlled for in all the included studies. The authors suggested that the reason for the slightly lower dislocation rate in the CoC-THA group than in the CoP-THA group maybe the use of a larger femoral head in the CoC-THA group than the CoP-THA group. This is consistent with THA biomechanics regarding increased jump distance with a larger head. Besides, a 10° or 20° lip liner on a standard polyethylene liner as well as a smaller head size may alter this outcome (*Bal et al., 2005*; *Lewis et al., 2010*).

Deep vein thrombosis has not been reported to be related to the implanted inlay. Although a meta-analysis revealed no evidence that material selection affects the risk of infection (*Hexter et al., 2018*), clinical studies in Australia and New Zealand showed that the risk of infection after CoC-THA was significantly reduced compared with CoP-THA (*Madanat et al., 2018*; *Pitto & Sedel, 2016*). This may be related to the reduction of bacterial biofilm formation caused by the chemical and physical properties of ceramic materials (*Holleyman et al., 2021*). However, it is important to note that before conducting clinical trials, factors such as BMI, comorbidities (*e.g.*, diabetes), nutritional status, and previous mobility status, which strongly influence infection rates, should be taken into account, as they can affect the final results of the trials.

The wear rate of traditional polyethylene is higher than that of ceramic, and friction can produce particles that lead to osteolysis (*Spinelli et al., 2009*; *Essner, Sutton & Wang, 2005*). However, the wear or osteolysis rate was non-significantly higher in the CoP-THA group. This may be because only three studies reported wear debris or osteolysis, and this limited series did not reach statistical significance. In many published series and joint arthroplasty registries, wear debris-induced osteolysis is the most common cause of postoperative revision surgery (*Pisecky et al., 2017*). In this meta-analysis, the revision rate was non-significantly higher in the CoP-THA group than in the CoC-THA group, just as rate of wear or osteolysis was slightly higher in the CoP group than in the CoC group. This can be attributed to the material properties of polyethylene, which is more prone to wear compared to ceramic. Polyethylene wear particles can induce a more pronounced inflammatory response, accelerating osteolysis and implant loosening (*Park et al., 2013*). The slightly higher wear rate in the CoP group highlights the importance of material selection in the long-term success of THA.

Young patients, typically defined as those under the age of 30, tend to pay more attention to revision surgery rates because they have longer life expectancies (*Shin & Moon, 2018*). However, the studies included in this meta-analysis involved patients >40 years old. It is possible that some patients who need revision surgery do not undergo it due to being deemed unfit for surgery due to medical conditions. There was no significant difference in the rate of common postoperative complications (dislocation, deep vein thrombosis, infection, wear debris or osteolysis) or in postoperative revision between the two groups, consistent with two previous meta-analyses (*Shang & Fang, 2021*; *Van Loon et al., 2022*). However, in addition to the bearing surface, other factors—such as the diameter of the femoral head (*Tsikandylakis et al., 2020*), the position of the acetabular cup (*Migaud et al.,*

*2016*), and the manufacturer of the prosthesis (*Zagra & Gallazzi, 2018*), and the surgical approach used (*Pincus et al., 2020*)—can also influence these rates, among others.

The postoperative Harris Hip Score was non-significantly higher in the CoC-THA group than in the CoP-THA group, and there was high heterogeneity among the studies. After excluding the main sources of heterogeneity, the postoperative Harris Hip Score was significantly higher in the CoC-THA group than in the CoP-THA group (WMD: 5.17; 95% CI [1.38–8.96]; $p = 0.007$). This contradicts a previous meta-analysis, which showed no difference in Harris Hip Score between the two groups (*López-López et al., 2017*). The specific reasons for the discrepancy need further exploration. However, in the study by *López-López et al. (2017)*, both the CoC and CoP groups used small heads and non-bone cement implants. Moreover, their focus was on comparing the Harris Hip Score of conventional metal-polyethylene (non-highly cross-linked), small-headed, cemented implants *versus* new materials implants. Therefore, we believe that our study, which specifically compares the postoperative Harris Hip Score of CoC and CoP bearings, is more targeted. Besides, although the HHS is subjective, patient-centered outcomes are more reliable since they reflect the patient's actual experience. Further clinical trials with longer follow-up are needed, as there are currently too few studies to support our findings (*Van Loon et al., 2021*; *Atrey et al., 2018*; *Ochs et al., 2007*; *Epinette & Manley, 2014*). In this meta-analysis, the preoperative and postoperative Harris Hip Score increased significantly in the CoC-THA group, compared to the CoP-THA group, in all the included studies. Patients may subjectively perceive less pain and better joint function after CoC-THA.

## Strengths and limitations

To the best of our knowledge, previous studies on the selection of bearing surfaces for THA are all from the perspective of doctors, focusing on the outcome of the operation. However, this current report is the first to focus on postoperative outcomes from the patient's perspective and to provide an important reference for patients in preoperative selection of bearing surfaces for THA.

Although we used the Cochrane collaborative-recommended GRADE system to evaluate the results, this study has several limitations. First, the clinical trials in the references we included did not take into account factors such as BMI, comorbidities, and they tended to use CoC bearings in younger patients, so these variables were not controlled well. Second, we analysed a small number of trials, precluding the generation of funnel plots to assess publication bias. The inclusion of studies with small sample sizes may lead to smaller study effects, leading to large standard deviations (SDs). Third, there were insufficient data to perform a subgroup analysis according to type of prosthesis, which may have introduced bias. Finally, since only two of the cited literatures used crosslinked polyethylene, there was insufficient data. We could not distinguish conventional polyethylene from crosslinked polyethylene. However, it is worth noting that in the analysis of wear debris, the use of highly cross-linked polyethylene liners was not included in the study, so the accuracy of the data is guaranteed.

## CONCLUSION

Pooled analyses demonstrated that CoC and CoP had comparable postoperative prognoses after initial THA. The overall and subtype analyses showed similar rates of dislocation, deep vein thrombosis, and infection. Although the results may not be statistically significant, CoC had better wear resistance, a lower osteolysis rate, and a slightly lower revision rate. After eliminating heterogeneity, the CoC bearing surface had a higher Harris Hip Score, and the prognosis was improved by CoC-THA. The findings could be pertinent to young patients such as those with hip disease due to avascular necrosis or dysplasia. However, the small number of studies included, and the presence of heterogeneity hamper the generalisation of our findings. Therefore, orthopaedic surgeons should select a THA material based on their experience and patient-specific factors, and large multicentre clinical trials with >15 years of follow-up are needed. Compared to conventional polyethylene, HXLPE is often regarded as a potentially better option due to its reduced wear debris. However, our data do not conclusively support that HXLPE reduces wear debris or affects postoperative complications more than CoC surfaces. This area should remain a primary focus for future research, with the hope that more orthopedic surgeons will take notice of this aspect and that more relevant articles will be published in the future.

### Funding

This work was supported by grants from the National Natural Science Foundation of China; Grant number: 82272489, 82203588; the TaiShan Scholars Project Special Fund; Grant number: NO.tsqn202306396; the Qingdao Traditional Chinese Medicine Science and Technology Project; Grant number: 2021-zyym28; and the Science and Technology Development Project of Shandong Geriatric Society; Grant number: LKJGG2021W082. The funders had no role in study design, data collection and analysis, decision to publish, or preparation of the manuscript.

### Grant Disclosures

The following grant information was disclosed by the authors:
National Natural Science Foundation of China: 82272489, 82203588.
TaiShan Scholars Project Special Fund: tsqn202306396.
Qingdao Traditional Chinese Medicine Science and Technology: 2021-zyym28.
Science and technology Development Project of Shandong Geriatric Society: LKJGG2021W082.

### Competing Interests

The authors declare there are no competing interests.

### Author Contributions

- Tingyu Wu conceived and designed the experiments, performed the experiments, analyzed the data, prepared figures and/or tables, authored or reviewed drafts of the article, and approved the final draft.

- Yaping Jiang conceived and designed the experiments, prepared figures and/or tables, authored or reviewed drafts of the article, and approved the final draft.
- Weipeng Shi performed the experiments, prepared figures and/or tables, and approved the final draft.
- Yingzhen Wang performed the experiments, prepared figures and/or tables, authored or reviewed drafts of the article, and approved the final draft.
- Tao Li conceived and designed the experiments, performed the experiments, analyzed the data, authored or reviewed drafts of the article, and approved the final draft.

## Data Availability

This is a systematic review and meta-analysis.

## Supplemental Information

Supplemental information for this article can be found online at http://dx.doi.org/10.7717/peerj.18139#supplemental-information.

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
