# Peer review of "Comparative postoperative prognosis of ceramic-on-ceramic and ceramic-on-polyethylene for total hip arthroplasty: an updated systematic review and meta-analysis"

_PeerJ, doi:10.7717/peerj.18139_

## Round 0.1 · original submission · Minor Revisions

This article has now been reviewed by 2 experts, and they are recommending that some revisions are needed before it can be Accepted. Therefore, please address their comments in a revision and resubmit.

Please note the comments from R2, that your PROSPERO registration needs to be clarified. Please be sure to respond to this issue

Specifically, the inclusion criteria need to be clarification regarding whether the THA you discussed includes primary THA or both primary and revision THA. This is important, because including revision THA will show less favorable outcomes and therefore skew the data and therefore compromise the analysis and the conclusion.

·

Basic reporting

I wish to appreciate the authors for their intrinsic efforts to make this efficient review article. As a reviewer i carefully reviewed this whole article. My comments listed below

1.The article well written and structured as per the guidelines.
2.It is consisted concise introduction so its should be more elaborate and discuss about the objectives, hypothesis in clear manner.
3. Figures quality will be improve before publication

Experimental design

No Comment

Validity of the findings

1.The wear rate of COP group higher than in the COC group (Lines : 212-215). This section need more elaborate discussion (How, Why)
2.The sufficient conclusion stated.

Additional comments

Some of the crucial statement stated without proper citation, so authors should be take essential action to solve this issue.
1. Please check lines 106-107, the sentence started by Of. If possible please rewrite this content.
2.Proper citation needed on Lines 118-119
3.Proper citation needed on Lines 158-161
4.Proper citation needed on Line 236
5. Some of the place, the author used the name like poly for polyethylene and some place full name used. please use full name over the manuscript.

·

Basic reporting

1. Language and Clarity:
- The manuscript is written in clear and professional English, with minimal grammatical errors.

2. Introduction and Background:
- The introduction provides a comprehensive background, establishing the context and importance of the study. It effectively identifies the knowledge gap and justifies the need for the current study.

3. Literature References:
- The literature is well-referenced, covering relevant studies that support the research. The authors have included a broad range of sources, ensuring a well-rounded background.

4. Structure and Conformance:
- The structure of the manuscript conforms to PeerJ standards. The sections are logically organized, facilitating easy navigation through the text.

5. Figures and Tables:
- Figures and tables are relevant, high-quality, and well-labeled.

6. Raw Data:
- Raw data is adequately supplied, following PeerJ policies.

Experimental design

1. Originality and Scope:
- The research represents original primary research within the journal's scope. The study addresses a well-defined research question, relevant to the field of total hip arthroplasty.

2. Research Question and Knowledge Gap:
- The research question is clearly stated, highlighting how the study fills an identified knowledge gap. The comparison of CoC and CoP bearing surfaces in THA is pertinent given the increasing use of these materials. Though the clarity on if only primary is included in the inclusion criteria or both revision and primary is included would be better.

3. Methodology:
- The methodology is described in detail, allowing replication. However, the inclusion criteria need clarification regarding whether THA discussed includes primary THA or both primary and revision THA.
- Suggestion: Clearly state if the study focuses only on primary THA or includes revision THA as well.
- Please clarify if the PROSPERO registration is valid since the data could not be confirmed.

4. Ethical Standards:
- Considering the study as a review article no concerns of ethics could be seen.

Validity of the findings

1. Data Robustness:
- The data provided is robust, statistically sound, and controlled except for PROSPERO registration.

2. Conclusions:
- The conclusions are well-stated and directly linked to the research question. They are supported by the results presented in the study.

3. Replication:
- Meaningful replication is encouraged. The rationale and benefit to literature are clearly stated, emphasizing the need for further research in this area.

Additional comments

Suggestions have been made as comments in the PDF enlcosed for reference.

---

## Round 0.2 · accepted · Accept

I am pleased to inform you that your amended manuscript "Comparative postoperative prognosis of ceramic-on-ceramic and ceramic-on-polyethylene for total hip arthroplasty: an updated systematic review and meta-analysis " is now recommended to be accepted and published. Congratulations. Thank you for publishing with PeerJ, we look forward to receiving future manuscripts from you and your colleagues. A/Prof Mike Climstein

·

Basic reporting

As per the reviewer suggestions, the manuscript has been corrected. All the queries addressed effectively according to the journal guidelines.

Experimental design

No Comments

Validity of the findings

All the finding modified as the reviewer direction.

Additional comments

The article may accept for publication.

·

Basic reporting

Basic reporting very much improved after the revisions made by the author and meets the standards well.

Experimental design

Clarification regarding the PROSPERO made the experimental design meeting the standards needed.

Validity of the findings

The findings in the study seems to be valid for the current scenario.